# Dynamics and roles in art and climate science collaborations: Experiences from the University of Hamburg

Anna Pagnone<sup>1,2</sup>, Jörn Behrens<sup>1,2,3</sup>, David Marcolino Nielsen<sup>1,2,4</sup>, Linda Jetter<sup>5</sup>, Lluc Vayreda Calbó<sup>6</sup>, Sam Burton-Weiss<sup>6</sup>, Dit Coesebrink<sup>7</sup>, Daniele Alef Grillo<sup>8</sup>, Carl Maria Kemper<sup>7</sup>, Nana Petzet<sup>7</sup>, Jenni Schurr<sup>7</sup>

15 Correspondence to: Anna Pagnone (anna.pagnone@uni-hamburg.de)

Abstract. Art and science collaborations enable new means for science communication, knowledge production, and activism. Previous work has often focused on the outcome of such endeavors and on an external description of these collaborations, and less on the personal dynamics in the collaboration itself. Our study of art and science collaborations in the project "Portraits of Climate" at the University of Hamburg takes a closer look at internal dynamics and roles and shows that the value or success of a collaboration does not depend on whether it is a one-way or a two-way collaboration, or whether the roles are classically separated in artist and scientist or are mixed. Instead, the decisive factors lie in mutual understanding, acceptance of team dynamics, and the subjective perceptions of participants – shaped by their motivations, backgrounds, and relationships with the partners – which often differs from an objective external description of the process observed. The project highlighted the importance of safe spaces, trust, and openness, while also revealing the fragility of these conditions without active facilitation and support. Although constrained to a small group of participants and visitors, the project demonstrated the potential of art and science collaborations to stimulate intellectual growth, to overcome one's predefined role, and to open a gateway for critical reflection, thereby catalyzing new approaches to addressing the environmental challenges of our time.

<sup>&</sup>lt;sup>1</sup>Cluster of Excellence Climate, Climatic Change, and Society, University of Hamburg, Hamburg, 20146, Germany

<sup>&</sup>lt;sup>2</sup>Center for Earth System Research and Sustainability (CEN)

<sup>&</sup>lt;sup>3</sup>Department of Mathematics, University of Hamburg, Hamburg, 20146, Germany

<sup>&</sup>lt;sup>4</sup>Max Planck Institute for Meteorology, Hamburg, 20146, Germany

<sup>10 &</sup>lt;sup>5</sup>Department of Geography, University of Hamburg, Hamburg, 20146, Germany

<sup>&</sup>lt;sup>6</sup>School of Integrated Climate and Earth System Sciences, University of Hamburg, Hamburg, 20146, Germany

<sup>&</sup>lt;sup>7</sup>Independent artist, Hamburg, Germany

<sup>&</sup>lt;sup>8</sup>Independent artist, Turin, Italy

# 1 Introduction

The historical interplay between art and science has evolved from a unified discipline into distinct fields, only to witness a convergence in the last decades through various forms of collaboration. This resurgence of art and science (or "SciArt") collaborations is driven not only by the need for new science communication formats (e.g., Hawkins and Kanngieser, 2017; Woodley et al., 2022) but also by the potential for novel forms of knowledge production (e.g., Kagan, 2015; Muller et al., 2015), emotional resonance (Muller et al., 2015), activism (e.g., Zhu and Goyal, 2019; Clark et al., 2020; Trott et al., 2020), and problem-solving policies (e.g., Benincasa and Eeckels, 2025). SciArt is enabling a deep epistemological shift toward integrated, transdisciplinary approaches in complex social and environmental challenges.

Many are the topics covered in the last years by SciArt collaborations, such as climate change (e.g., Rödder, 2017), rights of Nature (e.g., Greaves et al., 2025), desert ecosystems (e.g., Clark et al., 2020), citizen science (e.g., Stiller-Reeve and Naznin, 2018), oceanography (e.g., Jung et al., 2022), Antarctica (e.g., Stevens et al., 2019), and much more (Lanza, 2020).

Collaborations exist along a spectrum from primarily communicative projects, using artistic forms to translate scientific data, to projects pursuing the co-production of knowledge through interdisciplinary or transdisciplinary methods (e.g., Clark et al., 2020; Olazabal et al., 2024). Transdisciplinary work aims at transcending disciplinary boundaries, integrating diverse ways of knowing, to create shared frameworks and hybrid practices that generate new insights (Kagan, 2015; Clark et al., 2020).

Literature on SciArt collaborations in the context of climate change often focuses on outputs, on the additional creative and emotional approach compared to classical science communication (e.g., Woodley et al., 2022), and on the impact on the public, such as learning or engagement (e.g., Klöckner and Sommer, 2021). Some look at the process of collaboration itself, determining types of and phases in collaborations (Schauppenlehner-Kloyber and Penker, 2015; Clark et al., 2020; Gallois et al., 2024).

By moving beyond a focus on outputs, looking closely at collaboration processes and the people involved, this paper contributes to a deeper understanding of how art and science can work together to address climate change – specifically, on how exchange takes place, where the "co" in collaboration and co-creation sparks and which factors may hinder it. We illustrate the dynamics and participants' roles within five case studies of the project "Portraits of Climate" at the excellence cluster "Climate, Climatic Change, and Society" (CLICCS) of the University of Hamburg. Each case study was co-designed by an artist and one or more scientists and developed its own unique dynamic and internal structure. The presented case studies exemplify the fluid and dynamic nature of SciArt collaborations, along with the question of roles within collaborations.

# 2 Setting and case studies

The collaboration began in 2023 and lasted for one year. The collaboration took place in five teams: four of them composed by two people (one artist and one scientist) and one of them composed by three people (one artist and two scientists). Scientists span a wide range of seniority, from early "postdocs" to experienced professors, all based at the University of Hamburg. All scientists were volunteers in this project. The artist-scientist pairing was done by the lead author based on the participants'

interests. Some participants already had experience in SciArt collaborations; others did not. Each team produced a piece of art (Fig. 1), which was exhibited in the University Museum in Hamburg from November 7th, 2024, to August 10th, 2025. The project was supported by further colleagues at the University of Hamburg who were not part of one of the teams, such as technicians, communication officers, the museum director, and student assistants.

- A very open framework was intentionally chosen for the project. Only the time frame and the financial budget were set, with the simple requirement that each team creates a piece of art by the end of the collaboration year. Unlike some other SciArt collaborations, there was no prescription on how the collaboration should unfold, nor mandatory meetings or continuous oversight. Having a very open framework was a deliberate decision. The aim was to observe how collaboration between art and science evolves organically.
- While the five teams were free to meet as often as they wanted within the year, the large group met twice. One initial kick-off meeting served as a platform where all teams introduced each other and in the following mid-term meeting spaced 6 months, the teams presented their ideas and sketches, deepening everyone's understanding of what an SciArt project is.
  - The project was accompanied by semi-structured interviews of the participants at the beginning, at mid-term, and at the end of the collaboration period. These have been analyzed with respect to the dynamic and the roles' division in the collaboration.
- 75 The semi-structured interviews and participants observations lay at the base of the presented results.

# 2.1 Description of the case studies

Here we offer a short summary of the topics the teams decided to work on during the one-year collaboration:

Figure 1: The five pieces of art in the exhibition "Portraits of Climate" at the University Museum, Hamburg; (a) Way of No Return, (b) Mysterium Völlii, (c) Journey Through Time and into the Future, (d) The Little Shrimps are Running Out of Space, (e) Another Award. Credit: University of Hamburg / D. Masbaum.

(a) Way of No Return by Dit Coesebrink, Prof. Dr. Jörn Behrens, and Dr. Cleovi Mosuela

85

100

105

What connects art and science? The collaboration shows that art and science have fundamental principles in common: Both are exploratory, both experimental and both utilize the interplay between imagination and reality. However, the disciplines differ in what is perceived as beautiful about them - while art often plays with ambiguity, science emphasizes clarity and simplicity.

# (b) Mysterium Völlii by Nana Petzet and Prof. Dr. Michael Köhl

The Vollhöfner Forest has developed unnoticed over 50 years on flushing areas of harbor sediments in the Hamburg port area.

The forest now offers valuable insights into the gradual colonization of an area without any human intervention and the self-healing powers of nature.

# (c) Journey Through Time and into the Future by Jenni Schurr and Dr. David Nielsen

Climate change is accelerating the erosion of the Arctic coasts, which consist of frozen ground, the permafrost. A multimedia video performance addresses the dramatic changes, combining movement, sound, and visuals to illustrate the complex relationship between humans and nature, as well as the existential challenges posed by climate change. The work is based on Nielsen et al. (2024).

# (d) The Little Shrimps are Running Out of Space by Daniele Alef Grillo and Dr. Martin Döring

The North Sea Shrimps are intertwined with local traditions, fishermen's lives, and broader socio-ecological issues. The effects of globalization and climate change are reflected in the shrimps – a verbal portrait of a multilayered creature that has a place in society but is increasingly losing its space in nature.

# (e) Another Award by Carl Maria Kemper and Dr. Jan Wilkens

The work is about economic and political dependencies, extreme weather events and climate justice. How closely economic interests and ecological responsibility are linked can be seen when emissions-intensive industries are present at climate conferences, when affected communities demand climate justice and sue companies for compensation, and in local extreme events such as flooding (Wilkens and Datchoua-Tirvaudey, 2022; Aykut et al., 2022).

# 3 The collaboration between art and science

# 3.1 Dynamics in the collaborations

The analysis of the dynamics in the collaborations of the case studies is based on two aspects, the group and project development and the level of collaboration:

1. Group and project development – The five-stage-scheme of group development includes forming (orientation and familiarization), storming (conflict), norming (consensus and cohesion), performing (group work), adjourning (group dissolution). Schauppenlehner-Kloyber and Penker (2015) and Gallois et al. (2024) indicate that groups often don't follow this clear sequence of stages but go through iterative processes where stages overlap and blend into one another. In "Portraits of Climate", since many meetings occurred outside the pre-defined setting, not every stage could be directly observed. However,

- interviews indicate: a smooth forming stage; no real storming stage; a norming stage in which roles got clearer, and which also continued during the performing stage; a three-phase performing stage; an adjourning stage.
  - 2. Level of collaboration In literature, art and science collaborations are described along categories such as multidisciplinary, interdisciplinary, and transdisciplinary (Kagan, 2015; Clark et al., 2020). In "Portraits of Climate" we observe the whole spectrum of collaboration levels, from one-way to co-creative.
- Figure 2 describes the levels of collaboration (y-axis) of the five case studies along the three-phase performing stage (x-axis). The duration of the different phases, not represented in Fig. 2, varied between the teams. In the initial input and exchange phase ("exchanging" in Fig. 2), meetings laid the foundation, allowing participants to explain their work and discover common ground. The exchange between artists and scientists unfolded through a variety of formats, including informal gatherings like coffee meetups, studio visits, but also more structured in-depth interviews, email correspondence, regular online meetings, and joint field excursions. Conversations revolved around mutual presentations of each participant's field, work, and mindset. Methodologies from both the arts and sciences were explored and compared. The artists were particularly attentive during these interactions, engaging deeply with the scientific content to draw inspiration for their creative process. While some artists described this initial phase as a period of vivid and dynamic exchange, others experienced it more as a time of listening. One team embarked on a particularly integrated partnership, jointly seeking a new research question to explore together.








Figure 2: Collaboration levels throughout the project, divided into three phases: (1) input and exchange, (2) concept development, and (3) production. The y-axis shows the collaboration intensity, and the x-axis indicates the project timeline. Each colored line represents a team's collaboration, with red lightning showing an external disruption. Dotted lines highlight the discrepancy between participants' perceptions.

In the concept-development-phase ("developing" in Fig. 2), the creative process took on a more intentional and proactive character from the artists' side. They began to engage in discussions with their scientific partners, sharing sketches, and requesting specific materials to support the evolving work. Dynamic feedback loops emerged, as artists sought feedback on their progress, while scientists offered comments and suggestions that helped shape and refine the artistic direction. The intensity and frequency of feedback loops varied between teams, depending largely on the internal dynamics and working styles of the participants. For 3 out of 5 teams, this marked a transition from simply exchanging content to becoming actively involved in the creative process itself. It was during this stage that a shared understanding and mutual interest often solidified – although in one team, such a connection had been already established in the first phase but was not further pursued in the second phase.

In the final phase ("producing" in Fig. 2), the piece of art was produced. Three out of 5 artists produced the piece of art on their own because of their personal creative praxis, because they were forced by external conditions, or because they saw it as their responsibility (their role) in the team.

Following the lines of collaboration in Fig. 2, we see that no team stayed within one category (y-axis). In each team, the level of collaboration was fluidly shifting depending on the phase the team was in. This fluidity was also observed by Clark et al. (2020) in their SciArt collaboration. All in all, some scientists (3-4 out of 6) had limited involvement after the initial phase, while others (2-3 out of 6) maintained an active role throughout the project. One group really managed to keep up the collaboration even during the production phase, and the piece of art was co-produced till the end.

The collaboration was very much influenced by external factors (red lightning in Fig. 2), requiring adjustments and restrategizing. One participant went on paternal leave, limiting communication to email. One artist exchanged with several scientists before finding a partner (light blue line starting later in Fig. 2). Another artist, having initially two scientific partners, ended up with almost none – one dropped out soon; the other travelled a lot, making exchange difficult. Frustration made the artist almost drop out of the project. Another challenge was that the research object of one team was very dynamic: the initial plans of filming in the field could not be pursued due to unexpected ecological changes and the resulting worries of the local community.

While some collaborations remained limited to direct exchanges between the artist and their scientific partner(s), others expanded by incorporating expertise from additional sources. This included colleagues from different research fields within the university as well as external organizations like Deutsche Umwelthilfe. The integration of diverse perspectives proved to be particularly valuable for groups where the core collaboration was less constant, or the topic was very wide, so that further expertise was desired.







# 3.2 Roles in the collaborations

Literature describes both classical and mixed roles in art and science collaborations (e.g., Clark et al., 2020), and we observed both in the five case studies. Two teams had a more classical division of roles – "My role was one of a peer in a discussion process", with one scientist valuing the separation of roles – "I think that it's good that the scientist is not determining how the art is looking in itself". In these cases, developing the piece of art was the sole role of the artists. In another team, initially, the classical roles were respected. Over time, particularly before assembling the final piece, the roles mixed. The artist actively encouraged the scientist's involvement, moving beyond consultation to hands-on participation. As they met in the artist's studio – building the canvas installation, designing structures, selecting materials, and even engaging in stretching and dance – the scientist became a true co-creator. The next team started as a true transdisciplinary collaboration and mixed roles with a new research question to be addressed by both disciplines together and the artist wanting to do research. However, external conditions, miscommunication, and lack of commitment made the project turn to a one-way production. Due to limited availability on the scientific partner's side, the artist shared ideas during the development phase with the project coordinator. While this could be seen as a form of exchange, it caused confusion in the roles – the artist seemed to assume the coordinator was also a team partner, which was not the case. In the last team, the scientist felt that they and their partner escaped to prescribed roles by being "humans on an equal level with different backgrounds talking to each other and start trying to generate ideas." This human bond was also very important to overcome external challenges.

# 3.3 The "co" in collaboration, co-production, and co-creation

Besides the above objectively described level of collaboration, what strikes in Fig. 2 is that the level of collaboration was sometimes perceived differently by the members of the same team (dotted lines in Fig. 2). Thus, there is also a subjective level of collaboration. For example, 2-3 out of 6 scientists saw themselves as consultants and felt that the artwork was coproduced only in the "space of ideas". Meanwhile, the corresponding artists felt much more the collaborative nature of their piece of art. They viewed it as a shared effort from the start, emphasizing feedback loops and collaboration. The objective and subjective level of collaboration are strictly linked to the role an artist and a scientist take in the team.

Often, an enhanced value is given to transdisciplinary collaborations and mixed roles as enhancing coproduction. However, we argue that there is no reduced value in a one-way over a two-way collaboration or in mixed roles over classical roles, as long as the participants feel comfortable. What seems to be most important is the mutual understanding and acceptance of the teams' dynamics and the roles taken, no matter how these are.

We share two diverging examples. Although the dynamics and the roles in the developing and the producing phases between the two teams were seen as very similar for an external observer, the collaboration was valued very differently by the artists, one describing it as a collaboration and the other not.

In "Portraits of Climate" we observe that whether something is co-produced or co-created or not is not a mere objective description of the level of collaboration following defined categories, fluid or strict they may be. Valuing whether something








is co-produced or co-created or not has to include also the personal perception and thus the subjective level of collaboration. This is highly influenced by the participants' attitude, background, personality, experience, expectations — which ultimately go back to the motivation of taking part in the project — and by the relation they developed with their partner(s). Some participants were going beyond the roles of scientist and artist by being "more than the sum of the two", by being humans in relation and being humans with agency.

Considering this, we emphasize the importance of considering the perceptions of the participants, and of looking at how they overcome their roles to become simply humans, as opposed to scientists and artists constrained in their roles. This further expands Gallois et al. (2024), on their encouragement to transcend the persistent dichotomies of transdisciplinary collaborations, such as experts/non-experts, scientists/non-scientists, and artists/non-artists.

# 3.4 Impact on participants

In general, the participants found great enrichment in working with people from other disciplines, inspiring and learning from each other, and challenging their ways of thinking and engaging with climate change. The personal connections formed were very important. Most of the teams found it easy to collaborate because they had the sense of "sharing the same spirit".

For artists, working closely with scientists made the creative process more immersive and profound, they shared. Instead of starting from scratch, they could dive deeper and produce more targeted work with a clearer narrative, while allowing openness and structure at the same time. One artist appreciated that the rigor of scientific thinking helped order and define the direction in their artistic exploration. Another, initially skeptical about whether their medium could contribute "meaningfully", ultimately embraced the challenge of connecting abstract art with scientific ideas. An artist found only limited personal value in the project. Beyond some thematic inspiration and access to resources, the artist ultimately felt a sense of disappointment and frustration. The artist criticized the project's academic level and found the approach in the project naive. With many years of experience in the field, the participant brought expectations to the collaboration which were not satisfied. The extensive experience of the artist did not translate into a positive example of SciArt collaboration in "Portraits of Climate". Instead of evolving into a dynamic and inspiring collaboration, this exchange remained rather static and unfulfilling.

For scientists, the collaboration supported and further developed their overall attitude toward interdisciplinarity and transdisciplinarity and gave new perspectives. Three out of 6 scientists said that they thought more about how to communicate their science and about how to interact in creative ways with society. However, all but one scientist said that the collaboration did not influence their work. Once the artist worked with the material this one specific scientist provided, and came up with a story by themselves, the scientist noticed that they had focused on one aspect in their research, losing sight of another aspect the artist highlighted in the story (personal communication). This was a very refreshing moment which brought new momentum to the scientist's research.

For many participants, the experience reinforced the value of collaboration and the richness of exchanging perspectives. Knowing that such deep collaborations are possible – and understanding both their potential and limitations – was a key takeaway. Some noted that the collaboration confirmed what they had always believed – that combining different skills and

disciplines can lead to something truly valuable. Participants also valued the discussions – not only as knowledge exchange about art and science, but also about identity, relationships, perspectives, and ways of thinking. The conversations illuminated the potential of what lies "in-between" the disciplines. For all but one team, "Portraits of Climate" was educational and a process of intellectual growth. It also sparked reflections on skillfulness and human capability – how everyone, from scientists, to a cook preparing a sauce, possesses unique expertise that should be recognized and respected. This realization underscored the fundamental humanity of skill and craft, regardless of the field.

#### 3.5 Similarities







During the exchange, several participants (re)discovered similarities in aim and approach between art and science. In retrospective, most of the challenges the participants initially anticipated, such as finding a common language, did not materialize.

Literature largely looks at the boundary between art and science (in SciArt collaborations), however spanning from boundary as "great divide" (Halpern, 2011) to boundary as lines to be crossed (Kagan, 2015). In "Portraits of Climate" we shared the latter approach, looking at the two as "sibling disciplines", as one participant framed it. The ancient relationship between art and science, their nature, the approach and the methodologies used (gathering information, researching, experimenting, and testing hypotheses), and the way of inquiry, were recurring themes in the project. The group generally agrees on the following similarities between art and science: both fields examine "human consciousness and our being in the world"; they often draw inspiration from nature and represent complex realities; at the core of both art and science lies the act of questioning reality and the world we live in. In some interviews, artists explained that the aim of the artist is to expand the ways we see, feel, and express ourselves, pushing the boundaries of creative expression. Similarly, one scientist stated that science seeks to expand the space of knowledge. In this way, both art and science operate at the edges of their respective domains, striving to push the limits. This reinforces the idea that art and science, despite their differences, are parallel paths in the pursuit of knowledge and understanding (as in Kagan, 2015, and Benincasa and Eeckels, 2025).

One team found the parallels between art and science so compelling that their piece of art became an exploration of this very connection, rather than focusing on a specific research topic. Especially experimentation, abandoning ideas, and failure emerged as themes. All three are an essential and constructive part of the creative process in art as well as in science. Both recognized failures not as a setback but as an opportunity to push boundaries – challenging a society that is often fixated on performance and success. What happens in the moment of failure is often far more interesting than the original idea, leading to unexpected breakthroughs, so they agreed.

# 4 What is needed for a successful collaboration?

We agree with Woodley et al. (2022) that engaging in artistic practices can liberate researchers from rigid scientific communication frameworks, build confidence in new modes of expression, and should be valued by institutions and funders

alongside conventional scientific outputs. We add that what happens behind the scenes – in team dynamics, relationship building, and shared understanding – is equally critical for successful art and science collaborations.

#### 4.1 Enabling conditions for success

Research highlights several practical elements for structuring successful SciArt collaborations: creating safe spaces, competent project moderation, public recognition, organizational planning, and management of finances and time (Wright and Linney, 2006; Schauppenlehner-Kloyber and Penker, 2015; Clark et al., 2020; Schnugg and Song, 2020; Wright et al., 2023). However, before exploring the "how" of SciArt collaborations, it is important to first reflect on what is actually meant by a *successful* collaboration. Interviews from "Portraits of Climate" revealed that the most valued experiences were not necessarily tied to outcomes, but to process quality: feeling safe and trusted, sharing engagement in the creation of the piece of art – with an emphasis on a sense of togetherness rather than co-production in a strict sense, discovering unexpected insights, and overcoming challenges together. This aligns with Wright's (2023) finding that partnerships grounded in mutual respect are more likely to succeed.

Key enabling factors included safe space built on trust, honesty, and mutual respect, personal connections and personality fit,

open-mindedness, collaborative spirit, and curiosity, clear communication and shared understanding – yet these require
ongoing care, active facilitation, and structural support.

# 4.2 Challenges to collaboration



External barriers include limited accessibility, funding, timeframes, institutional support, and dissemination opportunities (Jacobs et al., 2017; Heras et al., 2021). Internal barriers include differences in language and expectations between art and science (Rödder, 2017; Gallois et al., 2024), power dynamics and "egos" (Gallois et al., 2024), and insufficient engagement with diverse knowledge systems (Steelman et al., 2018).

Two challenges were particularly evident in some teams or participants of "Portraits of Climate":

- 1. Communication Some participants repeatedly sought clarification on goals despite earlier discussions. One participant reverted to viewing the project as a "one-dimensional and instrumental arrangement" scientists producing knowledge, artists illustrating it contradicting earlier consensus on what we meant by co-creation in "Portraits of Climate". This highlights tensions in expectations and the difficulty of sustaining shared understanding over a year-long, fragmented collaboration, underscoring the need to unify diverse ways of knowing and communicating.
- Authorship From the outset, joint co-authorship was envisioned, so both parties felt represented in the work. This approach was maintained even when collaborations evolved in ways that made co-authorship questionable. Three participants addressed
   this directly: one scientist shifted from advisor to co-creator and embraced joint authorship; another felt their role was insufficient, though the artist disagreed, citing important contributions; one artist stressed that creative and conceptual development should remain the artist's domain, even in co-productions, to counteract the dilution of the artists' "aura of

singularity"; one artist saw co-authorship as a "huge problem" and naive without equal time, effort, and engagement in artistic thinking.

A cultural difference between art and science emerged in how co-authorship is recognized. In art, the notion of singular authorship often prevails, even when many individuals – technicians, collaborators, curators – have contributed. This can be seen in major works, from Michelangelo's *Sistine Chapel* to Christo and Jeanne-Claude's *Wrapped Reichstag*. By contrast, the scientific community routinely acknowledges multiple co-authors, reflecting varying levels of involvement. In some fields, such as particle physics, publications may even list hundreds of contributors.

#### 300 5 Further considerations

# 5.1 Open framework: Opportunity and challenge

A key challenge and opportunity of the project's open design was that teams had the freedom to define both topic, method, and collaboration style. This approach greatly influenced the development of each team's process and proved enabling for some, and disorienting for others, revealing both the potential and the limitations of this method.

Several teams made full use of the open structure. Groups that initially leaned toward a one-way production ended up evolving into highly collaborative and co-creative units. Others deliberately opted for a one-way collaboration or spread the collaboration to external contributors. Despite an occasionally challenging self-management of the groups, highly dependent on the group dynamics (Gallois et al., 2024), the case studies in "Portraits of Climate" demonstrated that, when given the freedom, a project can grow, transform, and flourish in unexpected ways.

One participant criticized the project for lacking structure and at the same time for not allowing enough freedom. The proposed approach ultimately matched the project's original intent: starting from scratch, developing shared interests, and co-producing outcomes. In hindsight, this team may have benefitted from larger support, maybe simply in the form of more frequent meetings with the whole group. Acting as "anchor points", these meetings could have further benefited the collective progress, ensuring a more regular engagement with the project, preventing misunderstandings, as well as actively helping the teams explore their diversity and establish their own group norms and goals (as in Cardenas and Rodegher, 2020).

# 5.2 The exhibition

The group's involvement in setting up the exhibition was entirely voluntary. Two artists served as curatorial assistants, all artists installed their own works, and only one scientist contributed to the process.

The exhibition collectively highlighted the variety of art and climate change and had a unifying purpose in the project. Yet frustrations with formalities and organizational issues – design, advertising, moderation, and content – detracted from its purpose.

Though not didactic, the exhibition fostered discussion and scientific explanation, ultimately connecting "Portraits of Climate" with science communication. In fact, though no instrumentalization was meant, most artworks carried also scientific messages.



Participants reflected that the exhibition served as a gateway to questioning and critical thinking, showing the potential of art and science collaboration to open doors. Three participant reflections underscored this point: by breaking down barriers between disciplines, the exhibition made both art and science more accessible; Still, its audience remained limited to exhibition-goers, restricting broader impact; From an activist standpoint, occupying diverse spaces remains essential for keeping climate discourse visible.

Visitor feedback, though not systematically recorded, revealed that the variety of scientific topics and artistic disciplines encouraged reflection, though some expected broader coverage of climate change given the exhibition's title. The art and science combination was seen as exciting, critical, and successful, though contextual scientific knowledge proved important.

# 5.3 The role of the lead author

The lead author, not part of any of the five teams, had multiple roles: coordinator, organizer, problem-solving reference, exhibition curator. This required close collaboration with the participants.

They also documented the process through interviews, but realized that their deep involvement in the project challenged complete objectivity. Some responses, while seemingly straightforward, had inconsistencies or nuances that complicated interpretation, knowing the full context of the collaboration. Personal connections between the lead author and the participants were crucial and positively influenced the project, yet when conflicts or misunderstandings arose, these also surfaced in interviews, making it hard to separate answers from personal issues with the lead author.

This dual role as collaborator and researcher added complexity to interpreting interviews but enriched response understanding.

Aware of potential bias, we aimed for a scientifically sound and balanced analysis.

#### **6 Conclusion**

Often faced by public inaction and skepticism, climate and environmental sciences have a great potential to benefit from societal policy engagement and dissemination. Through "Portraits of Climate" we ultimately fostered new means of knowledge production and outreach, inviting artists and scientists to freely collaborate. By doing so, we observed that combining different skills and disciplines can lead to collaboration that is more than the sum of artist and scientist – it can endow humans with agency. The most valued experiences were not tied to specific outputs, but to the feeling of engaging together in the creation of a piece of art.

We observed multiple and changing forms of collaboration between teams, and also within the teams themselves. The perception of contribution and ownership also varied substantially among teams and between team members, from co-authorship to external consultant and executor. At its best, the collaboration process enabled unexpected insights, fostered intellectual growth, and enabled the overcoming of challenges side by side. At the same time, we learned that these conditions are fragile. Safe spaces, mutual respect, and openness require active facilitation, clear agreements, sustained communication, and adequate structural support. Without these, the potential for intellectual growth is easily diminished.

https://doi.org/10.5194/egusphere-2025-5213 Preprint. Discussion started: 30 October 2025

© Author(s) 2025. CC BY 4.0 License.

While the project achieved visibility, including an illustration for the cover of *Nature Climate Change* as a side product of a 355 collaboration, its more enduring contribution may be the seeds that it planted for future transdisciplinary art and science collaborations addressing climate challenge and further challenges of the twenty-first century. Some participants were inspired to pursue new collaborative pathways beyond the project itself, but whether this momentum will last depends on whether we

and institutions can nurture such seeds.

**Author contributions** 

> AP wrote the project; AP, LVC and LJ managed the collaborations; JB, DMN, DC, DAG, CMK, NP, and JS participated in the collaborations; AP and LJ wrote the manuscript draft; JB, DMN, LVC, SB-W reviewed and edited the manuscript.

**Competing interests** 

The authors declare that they have no conflict of interest.

**Ethical statement** 

> The interviews for this paper were conducted through voluntary participation and with informed consent to share the outcomes in an academic publication. Besides being named as authors and coauthors of the artworks, participant identities and contributions have been anonymized in the text. All interviewees have been asked to give feedback and approval for publication.

Acknowledgements


AP, JB, DMN, LJ, LVC, and SB-W were funded by the Deutsche Forschungsgemeinschaft (DFG, German Research Foundation) under Germany's Excellence Strategy - 'CLICCS' (EXC 2037, Project Number. 390683824), contribution to the

Center for Earth System Research and Sustainability (CEN) of Universität Hamburg and to the Max Planck Institute for

Meteorology. The project "Portraits of Climate" was funded under the Excellence Strategy of the Federal Government and the

Länder. D. C., D. A. G., C. M. K., N. P., and J. S. were funded by the project "Portraits of Climate". We thank Martin Döring,

Michael Köhl, Cleovi Mosuela, and Jan Wilkens for participating in the project. We thank the Transfer Agency, the University

Museum and Antje Nagel, Stephanie Janssen, Remon Sadikni, and Laura Vogiatzis for their support. An AI tool was used to

improve some English formulations.

14

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
