# Peer review of "Dynamics and roles in art and climate science collaborations: Experiences from the University of Hamburg"

_EGUsphere, 2025_

## Referee Comment (RC1)

**Review Notes on Anna Pagnone et al. (2025) "Dynamics and roles in art and climate science collaborations: Experiences from the University of Hamburg."**

The following comments and suggestions are based on my professional experiences over the past forty-five years collaboratively working as a multimedia artist with scientists, engineers, and technology builders. This work involved facilitating innovation-oriented, ArtScience workshops for K-12 grade public and private schools, colleges, universities, cultural and research organizations, as well as Fortune 500 Companies and venture capital-funded technology startups who focused on fostering collaborative, innovative thinking (Siler, 1997, 2011; Seifter et al., 2008; Seifter, 2023). Many of these collaborations addressed climate-related science and global climate change challenges.

Note: I've provided some references and resources in my review of this manuscript (Pagnone et al., 2025). I hope the authors find them useful, as well as their readers.

Overall, I resonate with the goals, objectives, observations, and conclusions as stated in this manuscript. I appreciate the measured approach the authors have taken in organizing and assessing their exploratory project, "Portraits of Climate." Clearly, the case studies reveal that the collaborations 'foster growth, role expansion, and new ways to address environmental challenges.' That is, providing safe spaces have been established, along with a sense of trust and openness,' which the authors have elaborated on.

Also, I agree with the authors' general observations and perspectives on the 'decisive factors that determine the success of the dynamics and roles in art–science collaborations.' As they duly noted: 'the factors depend less on one- or two-way exchange, or fixed or blended artist/scientist roles, than on openness, trust, and mutual understanding'.

That said, here are four other decisive factors that are important to consider:

1. Defining or clarifying what constitutes "the work" of Art compared to the work [processes & products] of Science. Both share many similarities in perception, creative inquiry, discovery, process of intuition, insight and aesthetics (Siler, 1986). Many collaborators may not recognize the science *of* art and the science *in* art, anymore than they recognize the art *of* science and art *in* science. In these Postmodern times, "Art" can be seen *in* everything (e.g. "Design Is Everything"); it can used *in* everything from A to Z (Architecture to Zoology); and it can be utilized to connect everything, by all means and in all contexts (Siler, 1997).

Given the fact that we tend to experience things by how we define them, it's important to broadly define what we mean by "Art", "Science", and their interactions—as described and demonstrated by our examples. Each of these related, yet different, terms—Art & Science, art-science, Art/Science, Science/Art, SciArt, and ArtScience—implies something about the essence of *the work* and interactions, intentions, objectives, and outcomes (Siler, 1995).

2. Understanding the participants' subjective views on "the work" of Art and Science as far as they understand it. This gives collaborators the opportunity to share their motivations for collaborating and the outcomes they desire. Moreover, it helps the relationships with the partners, as it deepens everyone's understanding of one another.

3. Creating a better 'objective external description of the process observed'. To my knowledge, there is no standardized, definitive, external description that can bridge the gap between every participant's subjective views on the processes and products of Art compared to those associated with the process of Science applying the scientific method. We may need to design and test an objective external description technique to improve the evaluation and assessment process.

4. Developing constructive Feedback: What worked and didn't work in each of the five teams's collaboration. I've found that many of the problems participants in cross-disciplinary collaborations experience stem from overlooking or misunderstanding an individual's uniquely personal approach to art-making, "Art Think," and purposeful artistic productions. For many ArtScience practitioners today, art virtually connects all fields of disciplinary knowledge and processes of knowledge creation-and-sharing (Root-Bernstein et al., 2011; Siler, 1993, 1997 & 2014).

My overarching message to the authors: I think you've got the main ingredients for fostering creative and productive cross-disciplinary and transdiciplinary collaborations in which practicing artists and scientists create-and-share new knowledge in significant and commendable ways. You've documented in the five case studies your procedures and decisions for pairing the collaborators who represent various degrees of competences, expertise, experiences in these collaborations—duly noting, the senior participants in these collaborations are, as one might expect, more highly skilled and capable of ensuring the collaboration succeed in realizing a shared, common goal. Most importantly, your manuscript has a real *raison d'etre*—one which bears an inspired vision of some significant insights and promising offerings (i.e., breakthrough ideas, concepts, theories, patentable technological innovations, viable educational programs, etc.) that can be accomplished through these essential ArtScience collaborations.

Indeed, catalyzing and cultivating transdisciplinary collaborations in environmental sciences are more than necessary now: They're crucial for meeting the burgeoning, global climate challenges facing humanity today.

On that note: Consider the ways STEAM (Science, Technology, Engineering, Arts, Mathematics) knowledge, best practices and skills continue to contribute to the successes of businesses and industries involved in climate science, providing: Innovative Solutions (Siler and Ozin, 2012; Ozin and Siler, 2018; Ozin and Ye, 2024; Ozin and Ghoussoub, 2020; Seifter, 2023; Qian et al, 2015; Levy and Terranova, 2021); Data Analysis (ArtScience collaborations involving the creation of advanced algorithms and computational models that predict climate change impacts); Public Engagement and Awareness (Webinars such as "Art and Science Collaboration: The Key to a

Sustainable Future" organized by The Association for the Advancement of Sustainability in Higher Education [aashe] (November 1, 2017); Seifter et al., 2008, 2024; Lustig et al., 2025; communicating complex scientific data using storytelling, visual arts, generative AI to raise awareness about climate issues); Collaborative Research and Comprehensive Studies addressing climate challenges); Science & Policy Development (e.g., "Making science second nature." A project of the Denver Museum of Nature & Science); and Educational Initiative (Root-Bernstein, 2018, on STEAM education; STEMM, or Science, Technology, Engineering, Mathematics, and Medicine; Humanities, Arts, Crafts, and Design, HACD); Root-Bernstein et al., 2011).

Considering that the fact there are many definitions of Art and art-making in our Postmodern Contemporary Art, I would highly recommend that you include at least a few references that would inform your readers about the current developments that should neither be ignored nor overlooked. It would be like writing about the nature of human creativity, ingenuity and the process of applying "inventional & innovational wisdom"—on current climate science research and developments—without defining in advance what those terms and practices mean to the individuals, teams, groups, and organizations that are attempting to build practical innovations with this knowledge.

For example, over the past 14 years, I have been engaged in a productive ArtScience collaboration with the pioneer nanochemist, Professor Geoffrey A. Ozin, and his Solar Fuels Team at the University of Toronto. We've been exploring the possibilities of creating Artificial photosynthesis—with artworks and installations, titled "Nanoworld", exhibited at the international Armory Show in 2014 represented by Ronald Feldman Fine Arts in New York City. More to the point: Professor Ozin and I have been developing a new Periodic Table of Nanomaterials, which we envision using to create various technological innovations to help stem global climate change.

More to the point: Dr. Ozin has co-authored with his doctoral student, Mireille F. Ghoussoub, *The Story of CO2: Big Ideas for A Small Molecules* (Aevo UTP, 2020); this book "highlights technologies that can transform this molecule into valuable products while fighting climate change." Professor Ozin also co-authored with Jessica Ye, *The Story of Methane: Five Atoms that Changed the World* (Royal Society of Chemistry, 2024). Kindly consider the adventurous collaborative work he and his colleagues have done at the Karlsruhe Institute of Technology, Germany (https://www.kit.edu/kit/english/pi_2010_1849.php).

The purpose of mentioning these individuals and pointing out some relevant projects, initiatives, and enterprises is to expand and deepen the contents and contexts of Pagnone et al. (2025) manuscript. And, to provide some important guiding principles that can help foster timely cross-disciplinary or transdisciplinary collaborations.

It seems self-evident that artists and scientists alike are dealing with common human communications' challenges, which are further complicated by the compartmentalized knowledge and categorized data we're working with today (Siler and Ozin, 2024 & 2025). Perhaps, Artificial general intelligence (Agi) and Artificial superintelligence (ASI)

will help us integrate and learn to better communicate the works of art and science that are created to convey scientific concepts and visualizations to a general public.

Hopefully, Agi and ASI will prove to be helpful in making these ArtScience collaborations more accessible and relevant to the individuals collaborating as teams and groups, in which the obstacles to communication occur when the degrees of expertise and depth of experiences interfere. Maybe, these AI-enhanced technologies and systems will help decrease the complications that occur when collaborators encounter information outside their personal knowledge or explicit knowledge (Siler and Ozin, 2025 & 2024).

I gather from the interviews of the collaborators highlighted in the case studies, the individuals maintained a mostly cordial relationship that demonstrated their mutual respect and trust in one another's judgments, interpretations and appreciation of insights shared in the project "Portraits of Climate". That's a positive sign of progress!

I appreciate the authors's open-ended, rhetorical question and response [and I quote]: "What connects art and science? The collaboration shows that art and science have fundamental principles in common: Both are exploratory, both experimental and both utilize the interplay between imagination and reality. However, the disciplines differ in what is perceived as beautiful about them—while art often plays with ambiguity, science emphasizes clarity and simplicity" [end quote].

From my experiences, I have found that the scientific method welcomes the complexity of ambiguity in much the same way that the Arts & Humanities leverage metaphorical and symbolic languages. Consider the poignant statements made by Nobel laureates whose breakthrough discoveries and innovations were born from metaphors that inspired empirical studies. One example: the Nobel Prize-winning chemist and poet, Roald Hoffmann, who has written: "The images that scientists have as they do science are metaphorical. The imaginative faculty is set in motion by mental metaphor. Metaphor shifts the discourse, not gradually, but with a vengeance. You see what no one had seen before" (2006, 94, 407). I'd add, you also see a possible reality in the making with empirical research that either confirms it, or not.

My current ArtScience exhibition, "Metaphorming TIME" at the Madden Gallery, Museum of Outdoor Arts (https://moaonline.org/metaphorming-time) explores this evidence-based truth about artistic and scientific inquiries that describe how "metaphorms" are used to invent the future. This exhibition reaffirms a poignant observation by Fred Alan Wolf, Ph.D., author of *Parallel Universes* and *Taking the Quantum Leap*, has written: "It is time to realize that what science does best is create art, and what art does best is envision new science." I agree with Dr. Wolf's statement (and not just because he wrote this kind testimonial for my book, *Breaking The Mind Barrier,* Simon & Schuster, 1990).

One final, relevant note: "Meet the man building a starter kit for civilization."
You live in a house you designed and built yourself. You rely on the sun for power, heat your home with a woodstove, and farm your own fish and vegetables. The year is 2025," writes Tiffany Ng in The Download [on emerging technology] by Rhiannon

Williams, *The MIT Technology Review* 11.24.25.

   "This is the life of Marcin Jakubowski, the 53-year-old founder of Open Source Ecology, an open collaborative of engineers, producers, and builders developing what they call the Global Village Construction Set (GVCS)," continues Ms Ng. "It's a set of 50 machines—everything from a tractor to an oven to a circuit maker—that are capable of building civilization from scratch and can be reconfigured however you see fit. It's all part of his ethos that life-changing technology should be available to all, not controlled by a select few" [end quote].

This compelling article on Jakubowski' Open Source Ecology follows an earlier feature article, titled "Everything Is Design" (published in *MIT Technology Review* March/April 2023). Both articles point to some ways we can best move forward in creating and supporting programs that fully integrate Art-Science-Technology through ArtScience collaborations.

Ultimately, we may need to *converge* a world of divergent views on the processes & products resulting from these collaborations—and, provide a veritable means of measuring their efficacy.

I extend this "hopetimistic" (hopeful & optimistic) perspective, after reading Pagnone et al. (2025), which considers how transdiciplinary climate science collaborations can help improve the work of people and organizations aiming to solve our climate challenges.

**References and Resources**

Bennis, Warren, B., and Biederman, P. W. with Foreword by Charles Handy. (1997) *Organizing Genius: The Secrets of Creative Collaboration*. (Reading, MA: Addison-Wesley Publishing Company, Inc.)

Hoffmann, R. (2006). "The metaphor, unchained." *American Scientist*, 94(5), 407. https://doi.org/10.1511/ 2006.61.3496

Kuhn, R.L. (2025) https://www.newscientist.com/article/2498968-what-350-different-theories-of-consciousness-reveal-about-reality/
Kuhn, R.L. (2000) *Closer To Truth*, New York: McGraw-Hill (www.closertotruth.com).

Levy, Ellen K. and Terranova, Charissa N. [Eds.] (2021). *D'Arcy Wentworth Thompson's Generative Influences in Art, Design, and Architecture. From Forces To Forms.* Chapter 12: Todd Siler, "The Growth and Form of ArtNano Innovations: Inspirations from D'Arcy Wentworth Thompson's *On Growth and Form*" (London and New York: Bloomsbury Visual Arts); pp. 167-183.

Lustig, A.R., Crimmins, A.R., Snyder, M.O, Tanner, L., and Coller, I.V. (2025). "Bringing art and science together to address climate change," in *Climate Change*. 2025 Mar 6; 178(3):47. doi: 10.10007/s10584-025-03861-3

Ozin, G., and Siler, T. (2018). Catalyst: New materials discovery: Machine-enhanced human creativity. *Chem, 4*(6), 1183–1190. https://doi.org/10.1016/j.chempr.2018.05.011

Qian, C, Siler, T., Ozin, G.A. (2015). "Exploring the Possibilities and Limitations of a Nanomaterials Genome", *small* 2015, 11, No. 1, 64–69 Wiley-VCH Verlag GmbH & Co. KGaA, Weinheim.

Root-Bernstein, R., Siler, T., Brown, A., & Snelson, K. (2011). ArtScience: Integrative Collaboration to Create a Sustainable Future. *Leonardo*, 44(3), 192. https://doi.org/10.1162/ LEON_e_00161

Root-Bernstein, R. (2018) STEMM education should get "HACD". Incorporating humanities, arts, crafts, and design into curricula makes better scientists, 6 JULY 2018 • VOL 361 ISSUE 6397 ; http://science.sciencemag.org/ Published by the American Association for the Advancement of Science (AAAS).

Root-Bernstein, R.S. (1999). "The Sciences and Arts Share a Common Creative Aesthetic," in A.I. Tauber (Ed), *The Elusive Synthesis: Aesthetics and Science* (New York/Heidelberg: Kluwer Academic Publishers, 1996a), pp. 49–82.

Root-Bernstein, R.S. and Root-Bernstein, M. (1999). *Sparks of Genius: The 13 Thinking Tools of the World's Most Creative People*. (Boston: Houghton Mifflin Company, 1999); pp. 156-158.

Seifter, H. (2023). The Art of Science Learning. https://www.artofsciencelearning.org Seifter et al. (2008).The Art of Science Learning, Phases 1&2 of a National Science Foundation innovation initiative; http://www.artofsciencelearning.org/metaphorming/

Seifter, H., and Buswick, T. (Eds.) (2010). "Pointing Your Way To Success in Business Development," by Todd Siler in "Creatively Intelligent Companies and Leaders: Arts-based Learning for Business," *Journal of Business Strategy* (July-Aug, 2010): http://www.emeraldinsight.com/journals.htm?issn=0275-6668

Siler, T. and Ozin, G. (2025). Is the future of chemistry human and/or machine creativity? *Matter 8, Cell Press* 102220, July 2, 2025; pp.1-6 ; https://doi.org/10.1016/ j.matt.2025.102220. ©2025 Elsevier Inc.

___. (2024). There's no "next stop," only "full STEAM" ahead for humankind's AI future! *Cell Press*, Matter 7, December 4, 2024. https://doi.org/10.1016/j.matt.2024.11.001.

___. (2014) "NanoWorld," Ronald Feldman Fine Arts, at The Armory Show, New York, NY. (March 5 – 9); "Metaphorming Nature: ConnectingHuman/Nature's Creative Potential." Todd Siler in collaboration with Geoffrey Ozin, CU Art Museum, University of Colorado Boulder, Colorado (Sept. 6 - Dec. 20) (Catalogue)

___. (2012).''Cultivating artscience collaborations that generate innovations for improving the state of the world,'' SEAD:White Papers. Available at: https://seadnetwork.wordpress.com/white-paper-abstracts/final-white-papers/cultivating-artscience-collaborations-that-generate-innovations-for-improving-the-state-of-the-world/ *;* This material is based upon work supported by the National Science Foundation under Grant No.1142510, IIS, Human Centered Computing, "Collaborative Research: EAGER: Network for Science, Engineering, Arts and Design (NSEAD).

Siler, T. (2015). Neuroart: Picturing the neuroscience of intentional actions in art and science. *Frontiers in Human Neuroscience, 6*. https://doi.org/10.3389/fnhum.2015.00410

___. (2012). Neuro-impressions: Interpreting the nature of human creativity. *Frontiers in Human Neuroscience, 6*. https://doi.org/10.3389/fnhum.2012.00282

___. (2011). The ArtScience Program for Realizing Human Potential. *Leonardo*, 44(5), 417–424. https://doi.org/10.1162/LEON_a_00242

___. (2010). "The Mind And All It Creates," Fort Collins Museum of Contemporary Art, Fort Collins, CO, January 15 to March 25. (catalogue)

___. (2008-07). "All Representations of Thoughts for Art and Science," Tweed Museum of Art at the University of Minnesota-Duluth (December 4, 2007 to March 4, 2008).

___. (2007). "Todd Siler Adventures in ArtScience," at National Science Foundation, Arlington, Virginia (July 11 – November 9).

___. (2007). "Fractal Reactor: An Initial Computational Model for An Alternative Plasma Fusion System," in *Proceedings of the Fifth Symposium on Current Trends In International Fusion Research,* edited by Drs. Emilio Panarella and Charles D. Orth. NRC Research Press, National Research Council of Canada, Ottawa, ON KIA OR6 Canada). This Conference was organized under the auspices of the Global Foundation, Inc. and in cooperation with the International Atomic Energy Agency (IAEA), Lawrence Livermore National Laboratory, Los Alamos National Laboratory, Naval Research Laboratory, and Sandia National Laboratory.

___. (2007). "Fractal Reactor: Re-Creating the Sun," in *Leonardo Journal of Art, Sciences & Technology* Vol. 40, No. 3, pp. 270-278 (The MIT Press).
___. (2006) "Fractal Reactor: Re-Creating the Sun," Ronald Feldman Fine Arts, New York, NY (September 9 –October 7).

___. (2005). "Fractal Reactor: An Alternative Method and Apparatus for Plasma Fusion," in *Proceedings of the Fourth Symposium on Current Trends In International Fusion Research*, (pp. 411-426) edited by Dr. Emilio Panarella and Charles D. Orth. NRC Research Press, National Research Council of Canada, Ottawa, ON KIA OR6 Canada).

___. (2003). "Fractal Reactor: An Alternative Nuclear Fusion System Based on Nature's Geometry," [39-246] edited by Prof. Dr.-Ing. Sumer Sahin, June 03-08, Istanbul, Turkey; *ICENES 2007 Conference* was hosted by Gazi University, Ankara and Bahcesehir University, Istanbul.

___. (2001-02) "Thinking Utopia" (Utopisches Denken) article for the *Museutopia* Exhibition, Karl Ernst Osthaus-Museum, Hagen, Germany. *Museutopia* (June 11 - October 15) was presented conjunction with the Institute for Cultural Studies at the Center for Advanced Scientific Studies (Kulturwissenschaftliches Institut im Wissenschaftszentrum Nordrhein-Westfalen) in Essen.

___. (1997) *Think Like A Genius*, New York: Bantam Books; 304 pp. 198 illus, hardback edition; Transworld Books: January 1999; Ballantine Bantam Dell Publishers http://www.penguinrandomhouse.com/books/167113/think-like-a-genius-by-todd-siler
___. (1997). "Moving Beyond The Boundaries of Our Separate Worlds: The ArtScience of Learning, Creating, and Communicating," in *Borderless Thinking: Creating A Global Learning Society. 7th International Conference On THINKING*. Singapore International Convention and Exhibition Centre (June 1-6). Hosted by the National Institute of Education, Nanyang Technological University, Singapore.

___. (1995) "ArtScience: Integrating the Arts and Sciences To Connect Our World and Improve Communication," in *National Art Education Association*, 35th National Convention, Houston, Texas, April 11, 1995, p. 27-42.

___. (1993) "Artscience: Connecting Our World Through Metaphorms", *Science & Art: Creativity, Motivation, and the Joy of Learning* - a symposium hosted by The Chicago Academy of Sciences (October 28-31).

___. (1990). *Breaking the Mind Barrier: The Artscience of Neurocosmology*. (New York: Simon and Schuster); 416 pp. with 100 b/w line drawings, End Notes, Index, Bibliography; paperback, Touchstone Books, 1997); foreign editions: Chinese, Spanish.

___. (1988). *Metaphorms: Forms of Metaphor*. New York, NY: The New York  Academy of Sciences.

___. (1986). "Architectonics of Thought: A Symbolic Model of Neuropsychological Processes." Ph.D. inInterdisciplinary Studies in Psychology and Art, Massachusetts Institute of Technology. Available at: https://dspace.mit.edu/handle/1721.1/17200

___. (1982). "Everything," in *ALEA*, numero 3. Paris: Christian Bourgois Editeur, pp. 80-85. Article and drawings published in conjunction with the "Alea(s)" Exhibition at Musee D'Art Moderne De La Ville De Paris, organized by Jean-Christophe Bailly.